# Triple-Binder-Stabilized Marine Deposit Clay for Better Sustainability

**Mohamad Hanafi [1], Abdullah Ekinci [2,\*] and Ertug Aydin [1]**

[1] Civil Engineering, European University of Lefke, North Cyprus, TR-10 Mersin, Turkey; mhanafi@eul.edu.tr (M.H.); eraydin@eul.edu.tr (E.A.)

[2] Civil Engineering Program, Middle East Technical University, Northern Cyprus Campus, Kalkanli, Guzelyurt, North Cyprus, TR-10 Mersin, Turkey

\* Correspondence: ekincia@metu.edu.tr; Tel.: +90-542-888-1440

**Abstract:** Marine clay deposits are commonly found worldwide. Considering the cost of dumping and related environmental concerns, an alternative solution involving the reuse of soils that have poor conditions is crucial. In this research, the authors examined the strength, microstructure, and wet–dry resistance of triple-binder composites of marine-deposited clays and compiled a corresponding database. In order to evaluate the wetting–drying resistance of the laboratory-produced samples, the accumulated mass loss (ALM) was calculated. The use of slag alone as a binder, at any percentage, increased the ALM up to 2%. However, the use of lime as the third binder seemed to accelerate the chemical reactions associated with the hydration of clay and cementitious material and to enhance the chemical stability, i.e., specimens that included both lime and slag experienced the same ALM as specimens treated with cement only. Scanning electron microscopy analysis confirmed the durability improvements of these clays. The proposed unconfined compressive strength–porosity and accumulated mass loss relationship yielded practical approximation for the fine- and coarse-grained soils blended with up to three binders until 60 days of curing. The laboratory-produced mixes showed reduction of embodied energy and embodied carbon dioxide ($eCO_2$) emissions for the proposed models.

**Keywords:** cement; lime; copper slag; strength; durability; microstructure; $eCO_2$; embodied energy

## 1. Introduction

Marine clay deposits are found worldwide, especially in coastal regions. This research investigated the marine clays disseminated along the Mediterranean and northern coasts of Cyprus Island. Marine clays generate substantial construction problems, mainly due to their low strength and sensitivity against drying/wetting cycles. Rapid development of construction on those formations is comprised mostly of digging and dumping of soil. Environmental pollution created from digging and transportation affects mankind and the ecosystem. Additionally, the huge quantity of excavated soil affects the overall cost of shipping and controlling. Thus, companies try to manage the problems, allowing for environmentally friendly solutions. Considering the structural integrity, utilization, or reuse of untreated marine deposited clay for sub-base construction on highways poses engineering problems. The widely adopted solution is to dispose of the excavated soil into the nearest site or landfill. However, allowing for transportation between the quarrying plant and the excavation area can lead to a large amount of $CO_2$ emission and increased overall cost. The problems associated with those activities can be minimized via performance optimization of excavated soils, which might reduce the cost and negative effects on the environment. Every project requires different solutions for the management of excavated soil. Previous studies suggested alternative managing strategies for

excavated soils, including using them on-site, reusing excavated materials, pre-treating before use in construction, storing them for future consideration, and using them as landfill cover applications [1,2]. Furthermore, Magnusson et al. [3] reported that reusing excavated soil could save as much as 14 kg of carbon dioxide per ton. Additionally, Capobianco et al. [4] stated that the treatment of such soils is more beneficial than digging and dumping.

This study proposes the reuse of copper slag, which is extensively available at the Cyprus Mining Cooperation (CMC) site in the Lefke Region of Cyprus, as a cement replacement. The available slag is left over from the copper mining operations that ended in 1974 and is available in bulk form, having been haphazardly dumped around the Lefke Region. The serious concerns regarding the use of those materials involve heavy metal contamination and their leaching properties. Nevertheless, Alter [5] reported that the leaching values and heavy metal content of copper slag are lower than the levels prescribed by the United States Environmental Protection Agency (USEPA) and Basel Convention. Zain et al. [6] prepared specimens composed of 10% copper slag as a cement replacement. The results showed that the penetration of the trace elements did not exceed the normal rates. Another study revealed that the penetration of heavy metal (copper, nickel, lead, and zinc) ions from copper slag in large volumes was found to be lower compared with the prescribed limits from international authorities [7].

On the other hand, many researchers reported that copper slag does not show pozzolanic properties [8–10]. Moura et al. [11] studied the mechanical behavior of concrete containing 10% copper slag. The authors reported that compressive strength of concrete composed of copper slag had lower strength than reference concrete up to 91 days. However, other researchers mentioned that concrete incorporated with copper slag shows cementitious properties; furthermore, the pozzolanicity of copper slag increases the strength [12–16]. Additionally, another study assessed the pozzolanicity of clay composed of cement and copper slag [17]. The scanning electron microscopy (SEM) results revealed that composites exhibited pozzolanicity. The authors concluded that, at low cement content (30% dry weight of soil), the strength decreased with the copper slag amount. However, with an increase in the testing period and in the cement replacement level, the strength significantly improved with copper slag incorporation.

Few studies have considered the engineering properties of copper slag blended with clay and cement [18]. However, no studies have considered the durability properties of such composites. Durability is defined as the resistance to chemical attack, keeping its stability and integrity over a long period of exposure to a severe environment [19]. The durability of a silty clay–cement combination of composites was studied, where the authors found that the loss in mass of specimens decreased with increasing cement amount [20]. Furthermore, another research demonstrated that the lime addition could be decreased and have a positive effect on the ongoing wet–dry cycles over a long exposure [21]. Consoli et al. [22] and Consoli and Tomasi [23] investigated the porosity/cement and porosity/lime as durability parameters of soil composed of cement to evaluate the durability indices by considering the weight loss after several wet–dry cycles.

Many researchers investigated the lime stabilization of clay [18,19,21,24]. Choquette et al. [24], for example, examined the mineralogy and microstructure of lime-treated Canadian marine clays. Their results revealed that incorporating lime significantly caused the clustering of soil specimens. The flocculated arrangement was preserved by development of cementitious bonds between the particles. The authors correspondingly proposed the incorporation of calcium oxide (lime) into clay soil, which results in the formation of a plate-like morphology. The authors reported that this can increase the bulk volume of the small pores and space available between the clay–cement particles. Additionally, the authors stated that the modification in microstructure as a result of the lime addition agreed well with the mechanical properties of clay. Many researchers analyzed the cement–clay microstructural modification with SEM, reporting a decrease in the deflocculation level with a high amount of cement [25,26].

As specified in many standards around the world, unconfined compressive strength (UCS) is the most quantifying test in construction activities. The Australian earth-building handbook [27] suggests values for designing compressive strength between 0.40 and 0.60 N/mm² for rammed earth.

The New Zealand Standard for engineering design of earth buildings [28] uses a design compressive strength equal to 0.5 N/mm². In Bulletin 5 [29], a safe working compressive stress of 0.25 N/mm²—rather than an ultimate limit state—for stabilized rammed earth is recommended. Furthermore, according to the recommendations of the US Army Corps of Engineers (USACE) [30], the minimum required unconfined compressive strength for cement-stabilized soil for pavement base course is 500 psi (3400 kPa), and for the subbase, it is 250 psi (1700 kPa). Similarly, Maclean and Lewis [31] reported that the minimum required unconfined compressive strength for base and subbase designs of major roads is 500 psi (3400 kPa), and for minor roads, it is 250 psi (1700 kPa).

Treated soil deteriorates as a result of environmental conditions, such as wet–dry or freeze–thaw cycles and erosion. Under such conditions, the strength and stiffness values of the treated soils are reduced. Bonnot [32] stated that the durability of a treated soil can be studied in terms of loss of mass, expansion, change of strength, or swelling. As the current study was performed in Cyprus, which has a subtropical climate, i.e., Mediterranean and semi-arid-type climate, the wet and dry cycles are governing strength-control mechanisms. The US Army Corps of Engineers (USACE) technical manual [30] states that the maximum permissible mass loss of clay soils after 12 cycles (wet–dry) is 6% of the initial specimen weight for clay stabilization of pavements.

This study aims to evaluate the laboratory-produced triple-binder composites' strength, microstructure, resistance to wet–dry cycles, and sustainability performance. To address this research gap, in the current study, hydrated lime was incorporated with copper slag in cement-stabilized soil as a replacement to facilitate pozzolanic reactions. In addition, a porosity binder index for soil–cement mixes in terms of mass loss was examined for the first time, revealing a correlation between the mass loss and the unconfined compressive strength. Furthermore, embodied energy and embodied carbon dioxide (eCO₂) emissions of each mix were studied in terms of production and transportation of each stabilization product. This research could thereby enable a reduction in the amount of excavated soil through the assessment of the effects of soil disposal on the environment and on $CO_2$ emission caused by transportation. Additionally, incorporating various amounts of copper slag with cement could potentially decrease the impact of global warming and the amount of accumulated copper slag at the site.

## 2. Materials and Methods

### 2.1. Materials

The clay used in this study was collected from the project site, which was located in the Kyrenia District on the northern coast of Cyprus. The samples were obtained from basement excavation of a construction site. Basic characteristic tests, such as sieve analysis, specific gravity, and Atterberg limits, were evaluated based on the international standards in accordance with the corresponding ASTM D2487-17 [33]. The results of these tests are presented in Table 1. The clay was designated as inorganic low-to-medium-plastic clay (CL). The grain size distribution is shown in Figure 1. The soil is composed of clay, silt, and sand with percentages of 49%, 19%, and 32%, respectively. Furthermore, the X-ray fluorescence (XRF) test results showed that this clay is rich in $SiO_2$, $Al_2O_3$, and $CaO$.

**Table 1.** Physical properties of marine-deposited clay, hydrated lime, and copper slag.

| Properties | Marine Clay | Cement | Hydrated Lime | Copper Slag |
|---|---|---|---|---|
| Liquid limit (%) | 40 | - | - | - |
| Plastic limit (%) | 21 | - | - | - |
| Plasticity index (%) | 19 | - | - | Nonplastic |
| Specific gravity | 2.61 | 3.12 | 2.17 | 3.45 |
| Fine gravel (4.75 < diameter < 20 mm) (%) | 0 | 0 | 0 | 0 |
| Coarse sand (2.00 < diameter < 4.75 mm) (%) | 2 | 0 | 0 | 10 |
| Medium sand (0.425 < diameter < 2.00 mm) (%) | 3 | 0 | 0 | 82 |
| Fine sand (0.075 < diameter < 0.425 mm) (%) | 27 | 0 | 5 | 8 |
| Silt (0.002 < diameter < 0.075 mm) (%) | 19 | 90 | 90 | 0 |

| | | | | |
|---|---|---|---|---|
| Clay (diameter < 0.002 mm) (%) | 49 | 10 | 5 | 0 |
| Mean particle diameter (mm) | 0.0035 | 0.015 | 0.02 | 0.9 |
| USCS class | CL | ML | ML | SP |

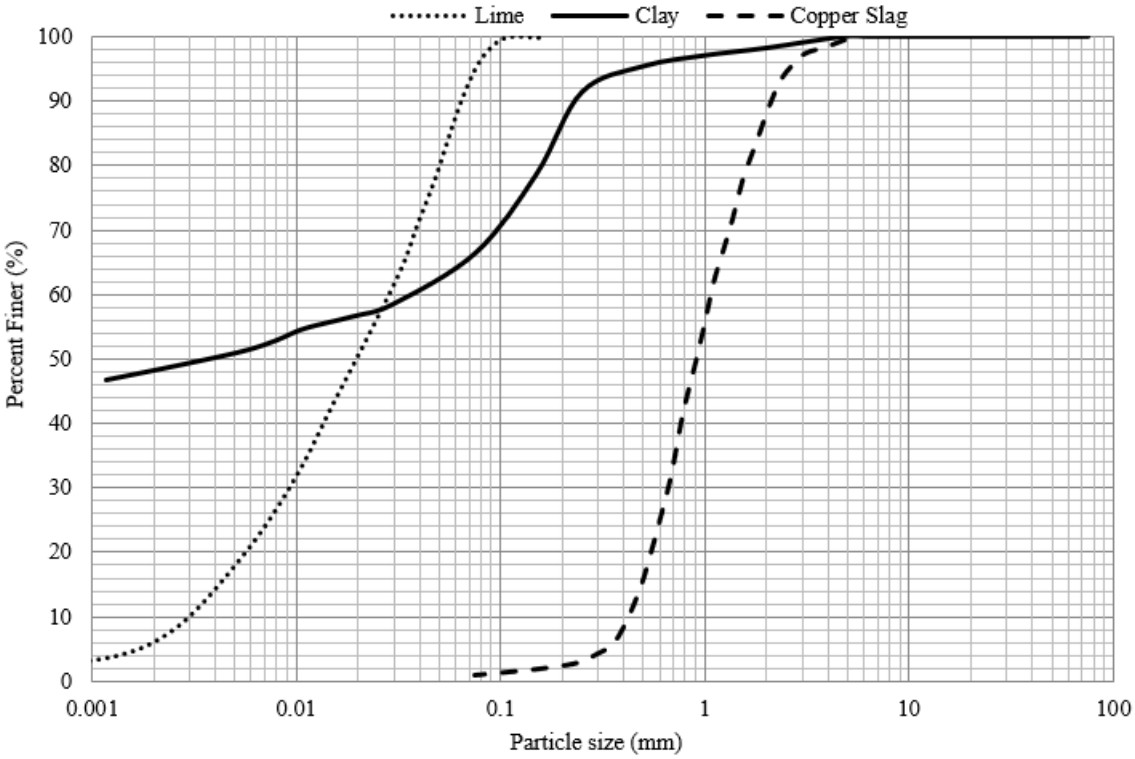

**Figure 1.** Grain size distribution of the studied clay, copper slag, and hydrated lime.

Copper slag was collected from an abandoned mine in the Lefke Region in Northern Cyprus. After performing the characterization tests, the slag was classified according to the Unified Soil Classification System USCS as poorly graded sand (SP), and its specific gravity was 3.45. The X-ray spectroscopy analysis allowed the determination of the main components of the slag, i.e., 43.5% ferrous oxide, 32.8% silicon oxide, 8.3% aluminum oxide, 4.0% CaO, and 2.6% SO₃.

Type I cement with a specific gravity of 3.12 and with a Blaine fineness of 289 m²/kg was used. The chemical composition of the cement is presented in Table 2.

**Table 2.** Chemical analysis of portland cement, hydrated lime, and copper slag. (EN197-1).

| Compound | Portland Cement (%) | Lime (%) | Copper Slag (%) |
|---|---|---|---|
| $SiO_2$ | 21.2 | - | 32.5 |
| $Al_2O_3$ | 5.1 | 0.38 | 8.3 |
| $Fe_2O_3$ | 2.5 | 0.3 | 43.5 |
| CaO | 64.7 | 70.89 | 4 |
| MgO | 0.9 | 1.95 | - |
| $K_2O$ | 0.2 | - | - |
| $SO_3$ | 1.5 | - | 2.6 |
| loss in ignition | 2.5 | 24.59 | - |

Hydrated lime contains mostly calcium oxide (71% CaO), and was obtained from a local supplier in Cyprus (imported from Turkey). The particle size distribution is shown in Figure 1. The physical properties of all used materials are presented in Table 1.

*2.2. Methods*

2.2.1. Molding and Curing of Specimens

To investigate the effects of clay treatment, cylindrical specimens of 50 mm diameter and 100 mm height were prepared. First, the amounts of materials were calculated from the targeted dry unit weight. They were then measured and dry-mixed in a tray with a flat-end spatula for at least 5 min to achieve uniformity. After that, water was introduced gradually while the mixing process continued. After ensuring the mixture's homogeneity, it was transferred to a split mold and statically compressed to achieve the required dry density. Upon completion of the mixing and compressing, the prepared specimens were transferred to a curing room in which they were kept for the required curing time [34]. The curing room had a 24 ± 2 °C temperature and a relative humidity of about 95%, according to ASTM C 511 [34]. The preparation data for all specimens are presented in Table 3.

**Table 3.** Details of molding and curing data.

| Name | Soil Type | Cement Contents (%) | Copper Slag Content (%) | Hydrated Lime Content (%) | Molding Dry Unit Weight (kN/m³) | Curing Periods (Days) | Test Type |
|---|---|---|---|---|---|---|---|
| Clay + Cement (CC) | | 7, 10 and 13 | - | - | 14.0, 16.0 | 7, 28, 60 | UCS, Wet-Dry Cycles * |
| Clay + Cement + Slag (CCS) | | 7 and 10 | 10% | - | 14.0, 16.0 | 7, 28, 60 | UCS, Wet-Dry Cycles * |
| Clay + Cement + Lime (CCL) | Marine Deposited Clay | 7 and 10 | - | 5% | 14.0, 16.0 | 7, 28, 60 | UCS, Wet-Dry Cycles * |
| Clay + Cement + Lime + Slag (CCLS) | | 7, 10 and 13 | 10% | 5% | 14.0, 16.0 | 7, 28, 60 | UCS, Wet-Dry Cycles *, SEM ** |

\* Wet–dry cycle done on all tested blends, 1.6 kN/m³ dry unit weights considering 28 days of curing.

\*\* Scanning electron microscopy (SEM) was done on untreated CCLS specimens, 1.6 kN/m³ dry unit weights considering seven, 28, and 60 days of curing.

Blending of the specimens was performed in relation to the relative constant of Portland cement (C), copper slag (CS), and hydrated lime (L). C is the mass of the cement divided by the mass of the dry clay; CS and L are defined as the quotient of mass of cement as a partial replacement for cement.

Porosity was calculated by using a modified version of Equation (1) proposed by Consoli et al. [35], dry unit weight ($\gamma_d$), and the weight contents of the marine clay ($W_S$), Portland cement ($W_C$), copper slag ($W_{CS}$), and hydrated lime ($W_L$). The corresponding unit weights are $\gamma_{ss}$, $\gamma_{sc}$, $\gamma_{sCS}$, and $\gamma_{sL}$, respectively.

$$\eta = 100 - 100 \left[ \left[ \frac{\gamma_d}{total\ mass\ of\ solids} \right] \right] \left[ \frac{W_S}{\gamma S_S} + \frac{W_C}{\gamma S_C} + \frac{W_{CS}}{\gamma S_{CS}} + \frac{W_L}{\gamma S_L} \right] \tag{1}$$

Depending on the cement porosity index ($\eta/C_{iv}$), a unique relationship was developed in order to predict the behavior of cement-treated soils [36], which only accounted for cement. More recently, Ekinci et al. [18] proposed a more general index $X_{iv}$, which accounts for all binder contents. In this study, Ekinci et al.'s [18] parameter was modified in an attempt to predict the strength for each mixture, where $X_{iv}$ was calculated from the modified Equation (2), where $V = W/\gamma_s$ is true for all of the used materials.

$$X_{iv} = \frac{V_S + V_C + V_{CS} + V_L}{V} = \frac{\left( \frac{W_S}{\gamma S_S} \right) + \left( \frac{W_C}{\gamma S_C} \right) + \left( \frac{W_{CS}}{\gamma S_{CS}} \right) + \left( \frac{W_L}{\gamma S_L} \right)}{V} \tag{2}$$

The external exponent of adjusted porosity/binder index $\eta/X_{iv}^{0.32}$ was determined to be the best-fit exponent for all of the blends studied herein and Ekinci et al. [18]. It is in accordance with previous empirical studies on various types of soils that obtained exponents that slightly varied between 0.28 and 0.35 [22,23].

Table 3 provides all of the necessary molding data, including material contents, curing periods, dry unit weights, and the types of tests conducted. The percentages of the cement used are related to the dry weight of the clay; for copper slag and hydrated lime, the percentages are related to the dry weight of the cement. In the triple blend, the copper slag is used as a replacement for cement, but lime is an addition of as much as the dry weight of the cement.

### 2.2.2. Compressive Strength Test

Strength tests were conducted on specimens after wetting and drying cycles. The tests were conducted according to ASTM C39 [37]. A fully automatic testing machine with 20 kN capacity was used. The failure load was recorded for every specimen, and an average of three specimens was used. Based on the procedure, if the single-specimen compressive strength deviated 10% from the average, the specimen was discarded, and a new specimen was prepared. Thus, the variation of the experimental results was completely eliminated.

### 2.2.3. Mass Loss by Wet–Dry Cycles

Durability tests were conducted according to ASTM D 559 [38] for durability testing of marine-deposited clays stabilized using various binders. These tests were used to evaluate the mass loss of composites through 12 wetting and drying cycles. Every cycle began with complete immersion of the specimens in water for 5 h; the specimens were then dried in an oven for two days. Subsequently, specimens were brushed with a wire brush using a pre-calibrated controlled load of about 15 N.

### 2.2.4. Microstructural Investigation

Scanning electron microscopy (SEM) was conducted to evaluate the influences of clay treatment with cement, slag, and lime on the microstructure at seven, 28, and 60 days curing. First, a small dried piece of specimen was attached to the aluminum stubs. Silver paint was applied. After that, the gold coating was applied. Magnifications of SEM images ranged from 3500x to 7500x.

### 2.2.5. Sustainability Investigation

Sustainability investigation was carried out to evaluate equivalent $eCO_2$ emission and embodied energy for the production and transport of used materials. It is worth mentioning that $eCO_2$ and embodied energy calculations were carried out based on their transportation distances to the Middle East Technical University, North Cyprus Campus, as a production site. Table 4 presents the embodied energy values and their equivalent $eCO_2$ emissions of the materials used in this research.

**Table 4.** Equivalent embodied carbon dioxide ($eCO_2$) emission and embodied energy for the production and transport of used materials.

| Process | Embodied Energy (MJ/kg) | $eCO_2$ Emission (kg $CO_2$/kg) |
|---|---|---|
| **Production** | | |
| Cement | 4.50 | 0.74 |
| Lime | 5.30 | 0.78 |
| Copper Slag | 1.60 | 0.083 |
| Water | 0.0009 | 0.000155 |
| **Transportation** | | |
| Cement through road | 0.35 | 0.32 |
| Cement through seaway | 0.0162 | 0.007 |
| Lime through road | 0.35 | 0.32 |
| Lime through seaway | 0.0162 | 0.007 |
| Copper slag through road | 0.0702 | 0.32 |
| Water | 0.016 | 0.32 |

In Northern Cyprus, most of the raw construction materials are provided from Turkey. The calculations were done in collaboration with the relevant supplier and available scientific resources. Therefore, when considering lime and cement $eCO_2$ and embodied energy values, the published data from the collaboration of Hammond and Jones [39] and the KASCON Group of Companies (Cyprus-based concrete ready-mix company) were used. Road transportation from the manufacturing place to the port and from the port to the production site was considered by using the European Parliamentary Research Service's (2018) publication for light-duty trucks. Seaway transportation emission and embodied energy values were obtained from the International Maritime Organization [40] and Rossit and Lawson [41]. Furthermore, $eCO_2$ emissions of the used water were obtained and calculated in collaboration with a governmental water resources agency. Additionally, copper slag values were obtained from Hammond and Jones [39].

The soil was assumed to be transported from the same site. Therefore, energy and emissions in excavating, transporting, mixing, and compacting were assumed to be constant for all mixtures and were not included in the total $eCO_2$ and embodied energy calculations. The total embodied energy and $eCO_2$ emissions that each mix will produce, along with the quantities of each material, were calculated and are presented in Table 5. In order to align with strength, durability, and microstructure studies, the mixes were all prepared at 1.6 kN/m³ density clay + 10% cement (CC), clay + 10% cement + 10% copper slag (CCS), clay + 10% cement + 5% lime (CCL), and clay + 10% cement + 5% lime + 10% copper slag (CCLS).

**Table 5.** Quantities of each material with $eCO_2$ emission and embodied energy calculations for each mix.

| Mix | Quantities (kg/m³) | | | | Embodied Energy (MJ/m³) | | | | eCO₂ Emission (kg CO₂/m³) | | | |
|---|---|---|---|---|---|---|---|---|---|---|---|---|
| | CC | CCS | CCL | CCLS | CC | CCS | CCL | CCLS | CC | CCS | CCL | CCLS |
| Production | | | | | | | | | | | | |
| Cement | 145.00 | 131.00 | 138.00 | 131.00 | 652.50 | 589.50 | 621.00 | 589.50 | 107.30 | 96.94 | 102.12 | 96.94 |
| Lime | 0.00 | 0.00 | 5.00 | 5.00 | 0.00 | 0.00 | 26.50 | 26.50 | 0.00 | 0.00 | 3.90 | 3.90 |
| Copper Slag | 0.00 | 16.00 | 0.00 | 16.00 | 0.00 | 25.60 | 0.00 | 25.60 | 0.00 | 1.33 | 0.00 | 1.33 |
| Water | 395.00 | 386.00 | 392.00 | 380.00 | 0.36 | 0.35 | 0.35 | 0.34 | 0.06 | 0.06 | 0.06 | 0.06 |
| Transportation | | | | | | | | | | | | |
| Cement through road | 145.00 | 131.00 | 138.00 | 131.00 | 50.75 | 45.85 | 48.30 | 45.85 | 46.40 | 41.92 | 44.16 | 41.92 |
| Cement through seaway | 145.00 | 131.00 | 138.00 | 131.00 | 2.35 | 2.12 | 2.24 | 2.12 | 1.02 | 0.92 | 0.97 | 0.92 |
| Lime through road | 0.00 | 0.00 | 5.00 | 5.00 | 0.00 | 0.00 | 1.75 | 1.75 | 0.00 | 0.00 | 1.60 | 1.60 |
| Lime through seaway | 0.00 | 0.00 | 5.00 | 5.00 | 0.00 | 0.00 | 0.08 | 0.08 | 0.00 | 0.00 | 0.04 | 0.04 |
| Copper slag road | 0.00 | 16.00 | 0.00 | 16.00 | 0.00 | 1.12 | 0.00 | 1.12 | 0.00 | 5.12 | 0.00 | 5.12 |
| Water | 395.00 | 386.00 | 392.00 | 380.00 | 6.32 | 6.18 | 6.27 | 6.08 | 126.40 | 123.52 | 125.44 | 121.60 |
| Total | | | | | 712.27 | 670.72 | 706.49 | 698.95 | 281.18 | 269.80 | 278.28 | 273.42 |

## 3. Results and Discussion

### 3.1. Compressive Strength and Porosity

Figure 2 presents the unconfined compressive strength ($q_u$) of specimens at 1.6 kN/m³ dry unit weight for all blends considering curing periods of seven, 28, and 60 days. In addition, 1.4 kN/m³ samples were also tested. However, as the results are, accordingly, similar to those of the 1.6 kN/m³ specimens, they are not presented in this section. Furthermore, the durability, microstructure, and sustainability assessments were also performed in accordance with 1.6 kN/m³ dry unit weight specimens; therefore, UCS results are presented accordingly to support the findings. Nevertheless, a table containing all of the results of UCS is presented in Supplementary Materials. Note that a more

comprehensive study on unconfined compressive strength of similar mixes with comprehensive statistical analysis has been published by Ekinci et al. [18].

Figure 2 reveals that, in all mixes, the increase of cement content and curing duration results in increase of compressive strength. It can be also seen that the CCS mix results in a slight reduction at all cement contents and ages. Adversely, the lime replacement of cement (CCL) mix appears to accelerate the hydration and results in maximizing compressive strength when compared with CC mixes at all cement contents and curing periods. It is also evident that the addition of lime to the CCS mix, which is CCLS, appears to contribute to the pozzolanic reaction; furthermore, as the curing period extends, the copper slag contribution becomes more evident. According to the US Army Corps of Engineers (USACE) technical manual [30] and MacLean and Lewis, the minimum compressive strength requirement for a road subbase construction of a major road can be satisfied via all proposed mixes. Similarly, for base course construction, it can be seen that the CCS mix at 7% cement content is the only mix that fails to satisfy the criteria.

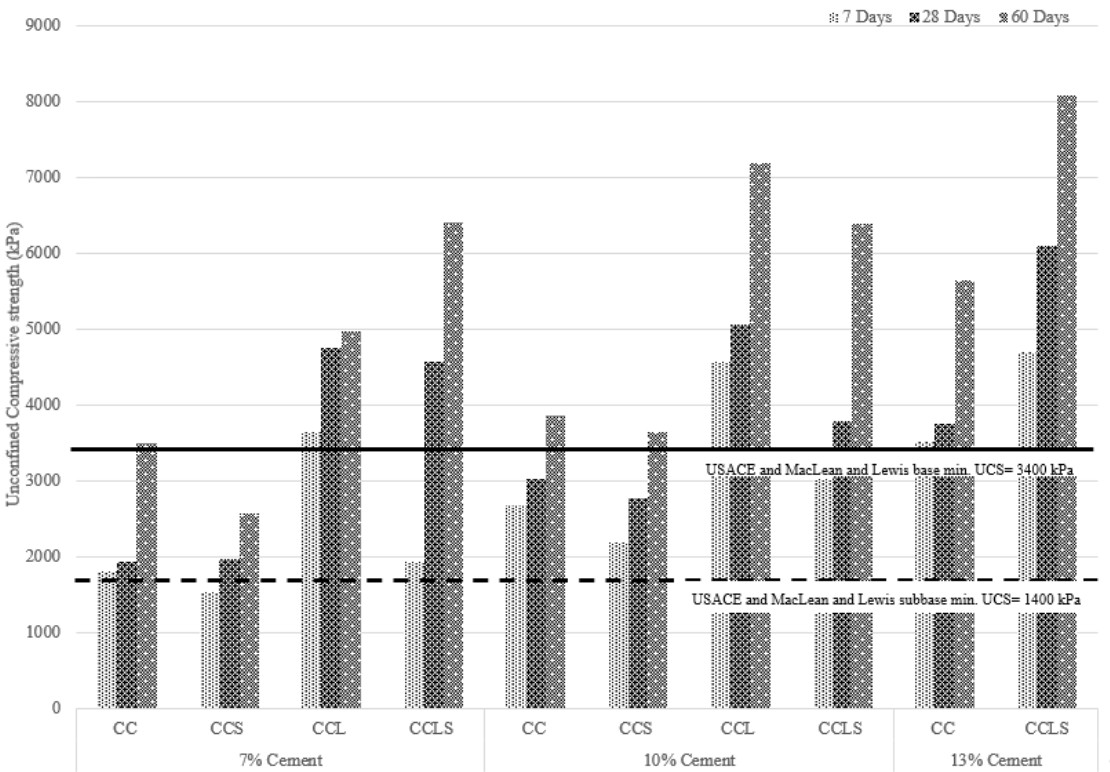

**Figure 2.** Unconfined compressive strength (UCS; $q_u$) at 1.6 kN/m$^3$ dry unit weight for all blends considering curing periods of 7, 28, and 60 days.

Figure 3 shows the relationship between the adjusted porosity/binder index against variation of cement content and curing period for all composite binders. It is clear that, for each blend, the increase of cement content results in reduction of the adjusted porosity/binder index. As with compression-strength observations, the CCS blend results in a slight reduction in all cement contents. It is interesting to note that the curing period did not affect the adjusted porosity binder/index. Nevertheless, it was only the CCLS specimen out of all other blends that showed reduction in the porosity/binder index as the curing period extended. This observation explains the reason for observed strength gain in CCLS specimens as the samples age. As expected, the lime replacement of cement CCL seems to accelerate the hydration and results in maximized reduction of the adjusted porosity binder index.

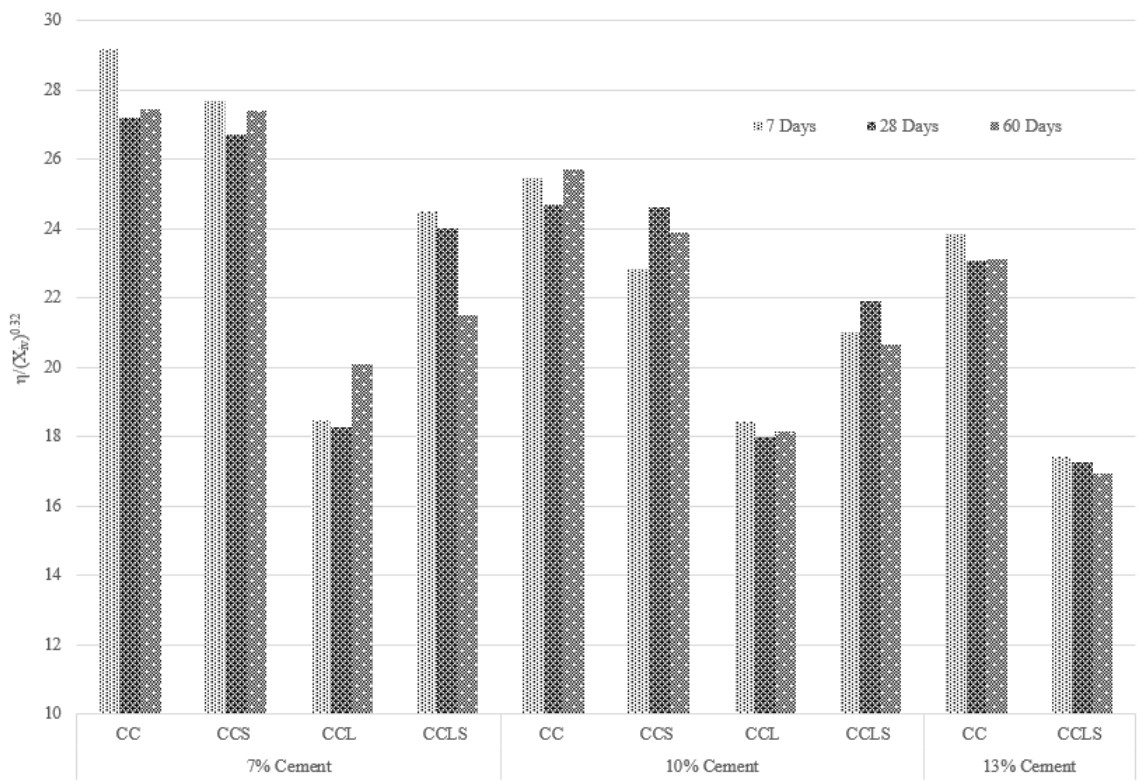

**Figure 3.** Adjusted porosity/binder index at 1.6 kN/m³ dry unit weight for all blends considering curing periods of 7, 28, and 60 days.

### 3.2. Mass Loss by Dry–Wet Cycles

Figure 4A illustrates accumulated mass losses (ALM) of marine-deposited clay soil–cement, soil–cement–copper slag, soil–cement–hydrated lime, and soil–cement–copper slag–hydrated lime blends after 12 wetting and drying cycles. The binder contents, distinct dry unit mass, and curing regime for the laboratory-produced specimens are summarized in Table 3. A sound polynomial fit of ALM versus $\eta/X_{iv}^{0.32}$ after durability tests could be obtained, as shown in Equation (3). Thus, it was observed that the adjusted porosity/binder index can be used to predict durability with up to a triple binder.

$$ALM = 0.031\left(\eta/X_{iv}^{0.32}\right)^2 + 0.864\left(\eta/X_{iv}^{0.32}\right) + 0.867, \ R^2 = 0.89 \tag{3}$$

The US Army Corps of Engineers (USACE) technical manual [30] states that the maximum permissible mass loss of clay soils after 12 cycles (wet–dry) is 6% of the initial specimen weight for soil stabilization of pavements. In this study, these wetting and drying requirements were satisfied for $\eta/X_{iv}^{0.32}$ of less than about 24. The shaded section in Figure 4A shows the data for the specimens that satisfied this requirement, which are all soil–cement–hydrated lime and soil–cement–copper slag–hydrated lime blends.

Finally, the unconfined compressive strength versus accumulated mass loss after 12 cycles is shown in Figure 4B for the above marine-deposited clays stabilized with clay soil–cement, soil–cement–copper slag, soil–cement–hydrated lime, and soil–cement–copper slag–hydrated lime blends. Unique second-order polynomial relationships with reasonable prediction can be obtained from Equation (4) for such blends.

$$ALM = 131.52(ALM\%)^2 - 2090.9\,(ALM\%) + 9265.5, \quad R^2 = 0.84 \tag{4}$$

Figure 4B also shows that the durability requirement of the USACE [30] for clay soils is only satisfied by the blends with unconfined compressive strengths greater than 1400 kPa. This finding

also defined a lower boundary for achieving satisfactory durability of such blends in terms of strength.

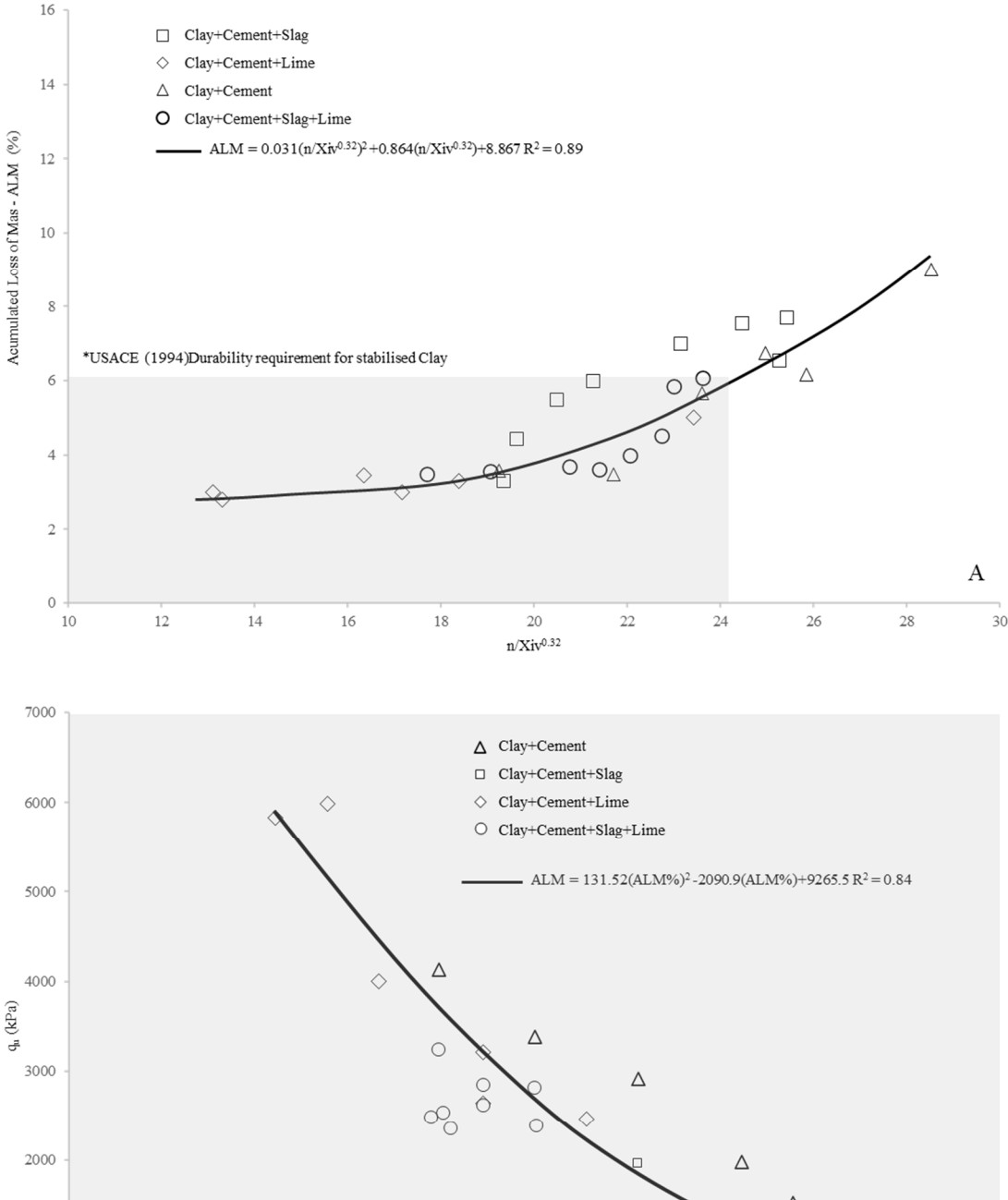

**Figure 4.** (**A**) Accumulated mass loss (ALM) versus adjusted porosity/binder index, (**B**) ALM considering twelve wet–dry cycles versus unconfined compressive strength (qu); for all tested blends, 1.6 kN/m³ dry unit weights considering 28 days of curing.

### 3.3. Microstructural Analysis (SEM)

SEM test results performed after compressive strength tests for the composites incorporated with marine-deposited clays at the 7, 28, and 60 days are shown in Figure 5. Those SEM results revealed that no cementitious bonds developed in the untreated clay due to the lack of soil-matrix (bond between the soil, aggregate, and cementitious compounds) cementing bonds. Additionally, as seen in Figure 5A–D, the tested marine-deposited clay is rich in calcium carbonate and contains "hollow-like structures".

The clay–cement mixtures were formed from the main chemical reaction of cement and the secondary reactions of the calcium-silicate-hydrate (CSH) product formation during the pozzolanic reaction of soil–cement mixtures. Figure 5B shows the products of hydration, needle-like crystals, and the products that resulted from calcium-silicate hydrates between the soil particles. Based on the SEM micrographs, the products had high aspect ratios. This confirms the formation of CSH needles in the bulk volume as a result of high strength and, thus, reduced ALM. Ettringite is a stable product with needle-like crystals with a hexagonal cross-section, which easily formed from specimens because of the high void ratios. This formation also caused the expansion and ALM increase at later time-points. However, it seems that the ettringite fills the pores in the matrix during the hardening period of seven to 60 days, and the pore space decreases significantly between those curing periods.

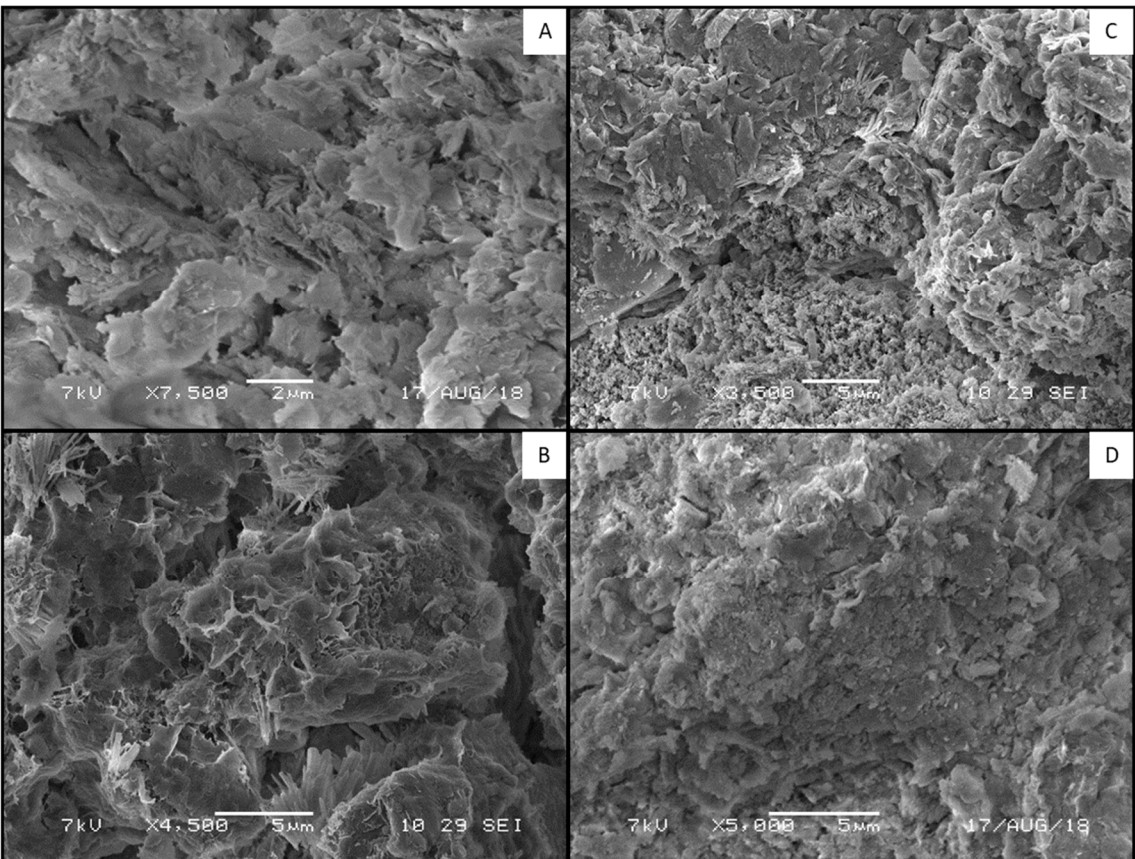

**Figure 5.** The scanning electron microscopy images, conducted on (**A**) untreated, (**B**) 7 days, (**C**) 28 days and (**D**) 60 days cured cement + copper slag and hydrated lime blend marine-deposited clays.

The clay compounds, silica, and alumina react with $Ca^{2+}$ and form CSH-calcium aluminate hydrate (CAH) during pozzolanic reactions. These hydrate products grow and harden, thus improving the ALM and strength of the clay-cementitious mixtures over hardening. After the 60 day curing period, the pores were filled with CSH gel, as shown in Figure 5C. Particle aggregation is observable due to the cation exchange reactions caused by introducing lime and slag. This aggregation reduces the "thickness of the double layer" between the clay particles and the attraction between particles, forcing the particles to move closer and initiating the particle aggregation phase.

As the curing period increased, the porosity of the specimens was reduced due to cementation, and a further improvement in ALM was observable. It can be assumed that day seven is the beginning of curing and the development of slag–lime reactions and corresponding pore spaces, as illustrated in Figure 4, where the porosities of the specimens cured for seven days are higher, and the porosity decreases with curing and with the addition of slag and lime. This can be seen in Figure 5C,D, which show the completed particle aggregation.

Figure 5D also shows the silica and alumina reaction, where the cementing property is more obvious. This is due to the secondary reaction development after 60 days, indicating a reduction in Portlandite (CH) due to the presence of copper slag and cement because of their reuse and transformation into secondary CSH at this stage of hydration. This feature is not evident in Figure 5B, as the reaction is in its early stage, and CH is more dominant because the CH crystals are absorbed in a later stage, since the copper slag reacts in a later stage. This characteristic is observable in the ALM reduction and strength development as well. Furthermore, hydrated lime addition appears to activate pozzolanic reactions at earlier stages, leading to a reduction in ALM.

### 3.4. Sustainability Assessment

The environmental assessments of the blends via embodied energy and $eCO_2$ evaluation are presented in Figures 6 and 7, respectively. Similar to Cabeza et al. [42], evaluation of both figures revealed a clear relationship between embodied energy and $CO_2$ footprint for primary production. Table 5 shows that greatest contribution to embodied energy and $CO_2$ emission is due to production and transportation of the cement, followed by the lime.

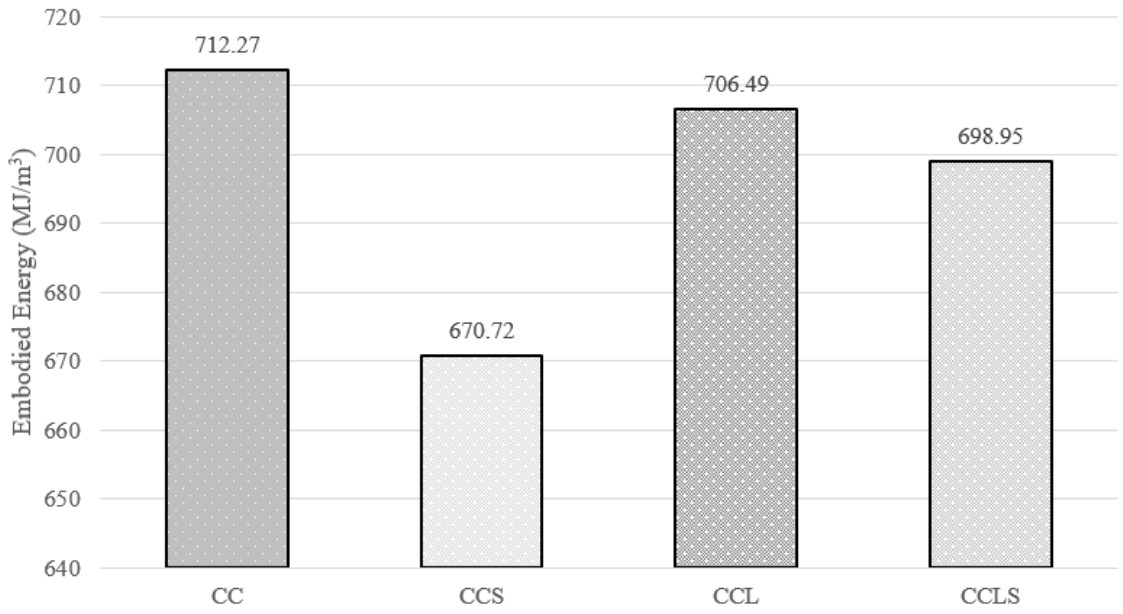

**Figure 6.** Embodied energy of 10% cement, 10% cement + 10% slag, 10% cement + 5% lime, and 10% cement + 5% lime + 10% slag mixes.

Figures 6 and 7 show that the CC mix, which can qualify as a control mix, produces the highest embodied energy (712.27 MJ/m³) and $eCO_2$ emission (281.18 kg $CO_2$/m³). Adversely, among all studied mixes, the CCS mix generates the lowest embodied energy (670.72 MJ/m³) and $eCO_2$ emission (269.80 kg $CO_2$/m³), which proposes at least a 5% reduction of environmental impact. Nevertheless, it is evident in the strength and durability analysis that, at early ages and low cement content, the CCS mixes' performance is degraded and does not satisfy the requirements. Therefore, as an alternative, a lime addition to the cement slag replacement mix (CCLS) was proposed. It can be seen that the CCLS mix generates the second-lowest embodied energy (698.95 MJ/m³) and $eCO_2$ emission (273.42 kg $CO_2$/m³), which proposes as much as a 3% reduction of environmental impact when

compared with the control mix (CC). Even though the replacement of cement with copper slag greatly reduces the impact, as demonstrated by the replacement of cement with lime (CCL), it contributes positively to strength and durability performance; however, the environmental impact reduction is not as effective, since embodied energy (706.49 MJ/m$^3$) and $eCO_2$ emission (278.28 kg $CO_2$/m$^3$) result in less than a 2% reduction.

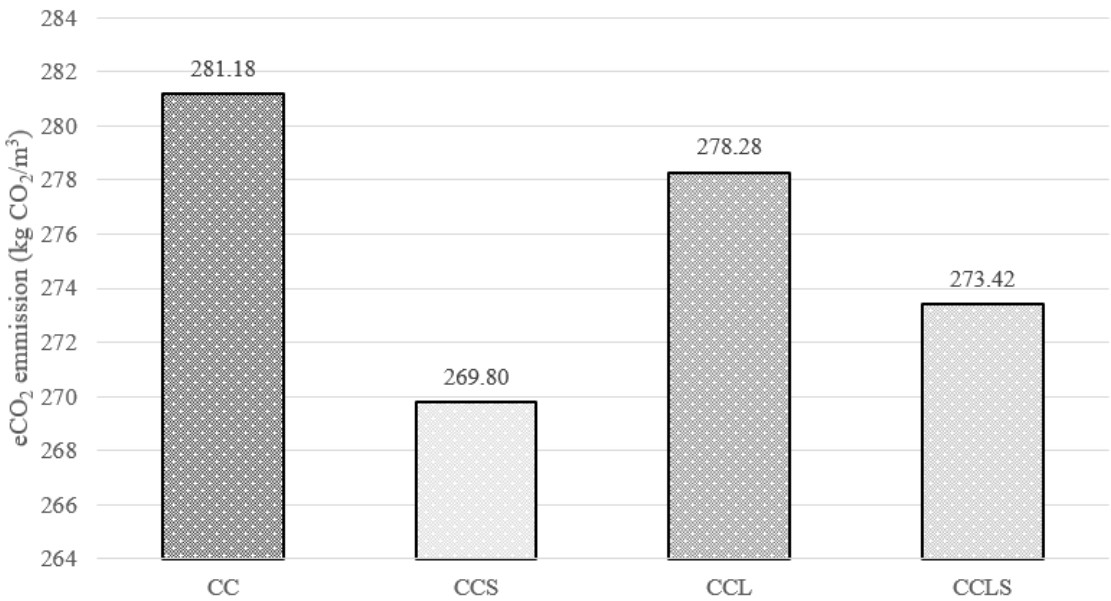

**Figure 7.** $eCO_2$ emission of 10% cement, 10% cement + 10% slag, 10% cement + 5% lime, and 10% cement + 5% lime + 10% slag mixes.

It is denotative from this study that the proposed binders reduce the consumption of cement and lime by increasing the amount of waste byproducts. The composites could be promising candidates in some major uses such as a base course, subbase course of major roads, rammed earth wall recommended design values, and structural fill. Additionally, they further reduce the environmental impact not just through the production and transport of the process, but also via disposal of waste. Furthermore, in a step ahead, Jiao et al. [43] presented the correlation between embodied energy and cost of individual building components. Therefore, use of this composite will also contribute to cost reduction.

## 4. Conclusions

This study examined the durability of the deposited marine clay when treated with cement, copper slag, and hydrated lime. The following can be concluded from the study:

- The incorporation of hydrated lime into cement: Slag-treated soils improved the strength and durability performance of the composites and ensured the satisfaction of weight loss and minimum compressive strength according to the requirements of USACE and MacLean and Lewis [31].
- As the curing period increased, the CCLS specimens' porosity declined due to pozzolanic reactions, and further improvement in ALM was observable.
- SEM pictures revealed the formation of "needle-like crystals" with a high aspect ratio between the particles, which resulted from the primary hydration. These crystals are responsible for the improvement in UCS and ALM.
- The incorporation of hydrated lime appeared to accelerate the pozzolanic reactions at earlier stages, resulting in a reduction in ALM.
- Environmental assessment of all proposed mixes resulted in the reduction of embodied energy and $eCO_2$ emission.

- Reusing unsuitable soil and hazardous wastes will reduce environmental and financial impacts. Improving soil with additives will facilitate the use of the available soil on site. In addition to the environmental contribution of cement usage reduction, using waste material, such as copper slag, will enable safe disposal of those harmful materials.

## 5. Recommendations

In this study, a formula to predict the accumulated loss of mass when adding a triple binder was successfully created. Further research can be conducted to validate this formula when using pozzolanic materials other than copper slag.

**Supplementary Materials:** The following are available online at www.mdpi.com/2071-1050/12/11/4633/s1, Table S1: Unconfined compressive strength results of all performed tests.

**Author Contributions:** M.H. and A.E. conceived the study and were responsible for scheduling and performing the experimental study. E.A. was responsible for microstructure interpretation. A.E. and M.H. wrote the first draft of the article, and E.A. oversaw and finalized the study. All authors have read and agreed to the published version of the manuscript.

**Funding:** This research was funded by Office of Research Coordination, Middle East Technical University, Northern Cyprus Campus, grant number FEN-20-YG-4 and The APC was funded by FEN-20-YG-4.

**Acknowledgments:** The authors greatly appreciate the discussions and help from Pedro Ferreira from University College London and Nilo C. Consoli from Universidade de Federal do Rio Grande do Sul. The authors also thank the graduate students Burak Kın and Dogan Gülaboglu for their support during the laboratory experiments.

**Conflicts of Interest:** The authors declare no conflict of interest.

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
