# Peer review of "Triple-Binder-Stabilized Marine Deposit Clay for Better Sustainability"

_sustainability, doi:10.3390/su12114633_

Round 1

Reviewer 1 Report

Title: Durability and microstructural properties of triple-binder-stabilized marine deposit clay

General comments:

The manuscript deals with marine clay stabilized with a mixture of binder
materials in laboratory. The study is interesting, but there is no justification, why for example these testing methods were chosen. For the use of pavement materials some other testing would be needed depending on the idea, where and how the material is used. The environmental acceptability
and CO2-emission calculations are not presented. This is OK, but adding them would improve the content.

The language is good and only small things should be improved.

A relatively big amount of the samples were tested, but I would like to see more analysis of the results. For example curing time – UCS strength figure would be useful as well. The objective of the study should be stated more clearly. An explanation is needed which binders are actually used, where the clay comes from? If the clay is stabilized, how it is used afterwards?

Detailed comments:

Row 65 check language “… an uncreased the testing…”

Row 101-102, Why ASTM standards are used? not ISO or EN? The research is done in European country.

Row 112. “…allowed us to…” Why you have used “we”? Sounds a bit odd here.

Row 117. Mention that all material properties are presented in Table is not correct. At least cement is missing.

Row 122. How the mixing was done?

In rows 136 - 141, how many samples were prepared for UCS tests? And for other test types?

Figure 2. Why you have named the samples as “soil”? “Clay+…” would be better.

Figure 2b. Check how the ALM and UCS strength requirements are presented. Maybe do as in 2a and add a raster to acceptable area, which is above 1400 kPa, not below as it could easily be guessed.

Row 174. 3.2 Durability tests, should this be something else? 3.1 is already Durability tests.

Author Response

We would like to thank to reviewer #1 for such extensive reviews. According to the feedbacks from reviewer #1, the manuscript has gone through a rigorous amendment.

Reviewer 2 Report

This manuscript aims to use cement, copper slag and lime to stabilize marine deposit clay. The motivation towards sustainability in construction is very valid. Some experimental results were reported and analyzed. However, some critical issues are identified for this manuscript:
1.       Durability covers many aspects like leaching, sulfate resistance, cracking, etc. This manuscript only reports the accumulated mass loss to represent durability. I think the title should be more specific in AML rather than durability.
2.       SEM is a powerful tool to show internal structure at micro-level. However, the discussion part of this manuscript is totally based on four SEM images. This seems to be suspicious because no other evidence is provided, such as the porosity.
3.       The particle size of copper slag is very big according to Fig. 1. This strongly affects the activity of slag. But this manuscript does not really provide sufficient scientific explanations.
4.       Section 3.2 should change the subhead name.
5.       Conclusion 1 is not supported by the experimental data.

Author Response

We would like to thank to the reviewer for such extensive reviews. According to the feedbacks from the reviewer, the manuscript has gone through a rigorous amendment.

Reviewer 3 Report

The paper addresses the unconfined compression strength, the accumulated loss mass during wetting and drying cycles as well as microstructural analysis by scanning electron micrographs of a mixture of clay, Portland cement, cooper slag and hydrated lime.

Although the mixture is somewhat innovative it was not completely clear to me, why do we need those four ingredients. From literature, lime is generally the most adequate binder for soft clayey soils due to flocculation of clay particles and water content reduction. This seems to be in agreement with Figure 2 results. However, the addition of slag does not improve strength and the accumulated loss mass is not reduced either. So, I don’t understand the focus of the paper on this triple binder. If the idea is to reduce the consumption of cement and lime and increase the amount of waste by products (such as slag) this needs to be clearly explained. Anyway, I would like to see the results of soil+lime mixtures even increasing the lime content if necessary (5% seems rather low).

Does Figure 2 includes all results, namely at different curing ages? How the curing time affects the n/Xiv ratio? It should be clearly explained what is Xiv in this ratio and how the exponent 0.32 was obtained.

In my opinion, the abstract should be revised because it gives a very good summary for someone that knows the paper but not for someone that it is reading this for the first time. For instance, the reader does not know that the ALM was on wetting and drying cycles.

There are other minor issues that could be improved:

  • Keywords: X-ray techniques – X-ray spectroscopy was used for the slag chemical analysis, but this doesn’t seem a very important point of the paper.
  • It should be highlighted that the marine clay is soft
  • Some English problems [line 27 – are found; line 28 – along the northern Mediterranean coast; line 57 – does not show; line 89 – binder; line 140 – the sample was discarded]
  • Line 41 – do you really mean quarried soils, referring to the clay ??
  • line 51 – have reported that the leaching values of these materials are lower than the levels prescribed …. ???
  • line 59 – had lower strength than reference concrete
  • line 65 – with an increase in the testing period and in the cement replacement level, the strength ….
  • Line 70 – Durability is defined as the resistance to chemical attack keeping its stability and integrity….
  • Line 78 – lime treated clays is a very well studied topic for just citing this single reference. [24]
  • Line 126 – temperature and humidity of the curing room could be given
  • The word sample should be replaced by specimen since sample is for natural ones retrieved in situ while specimens is for artificial ones prepared at the lab.
  • Line 241 – Universidade

Author Response

We would like to thank to the reviewer for such extensive reviews. According to the feedbacks from the reviewer , the manuscript has gone through a rigorous amendment.

Round 2

Reviewer 2 Report

As suggested in the previous comments, the durability in this study actually only refers to the mass loss of wet-dry cycles. Hence, the name of Section 2.2.3 and Section 3.2 should be changed to be more specific.

Author Response

We would like to thank reviewer #2 for reviews. We believe that without motivation, support, and excellent contribution of reviewer #2, we are not able to reach this final version. Please accept our sincere thanks.

According to the suggestion of reviewer #2, section 2.2.3 (p.6 line 195) and section 3.2 (p.9 line 276) title has been revised accordingly as “Mass loss by dry-wet cycles” 

Reviewer 3 Report

Thanks for your improvements.

Author Response

We would like to thank the reviewer for provided feedbacks. We believe that without motivation, support, and excellent contribution of the reviewer, we are not able to reach this final version. Please accept our sincere thanks.